# Unravelling the Triad of Lung Cancer, Drug Resistance, and Metabolic Pathways

**DOI:** 10.3390/diseases12050093

**Published:** 2024-05-06

**Authors:** Pratik Mohanty, Babita Pande, Rakesh Acharya, L V K S Bhaskar, Henu Kumar Verma

**Affiliations:** 1Department of Bioscience and Bioengineering, Indian Institute of Technology, Guwahati 781039, India; pratikmo122@gmail.com; 2Department of Physiology, All India Institute of Medical Science, Raipur 492099, India; babitatime2014@gmail.com; 3Department of Zoology, Guru Ghasidas Vishwavidyalaya, Bilaspur 495009, India; rakeshacharya090@gmail.com (R.A.); lvksbhaskar@gmail.com (L.V.K.S.B.); 4Lung Health and Immunity, Helmholtz Zentrum Munich, IngolstädterLandstraße 1, 85764 Oberschleißheim, 85764 Munich, Bayren, Germany

**Keywords:** lung cancer, metabolism, signaling pathway, drug resistance, disease management

## Abstract

Lung cancer, characterized by its heterogeneity, presents a significant challenge in therapeutic management, primarily due to the development of resistance to conventional drugs. This resistance is often compounded by the tumor’s ability to reprogram its metabolic pathways, a survival strategy that enables cancer cells to thrive in adverse conditions. This review article explores the complex link between drug resistance and metabolic reprogramming in lung cancer, offering a detailed analysis of the molecular mechanisms and treatment strategies. It emphasizes the interplay between drug resistance and changes in metabolic pathways, crucial for developing effective lung cancer therapies. This review examines the impact of current treatments on metabolic pathways and the significance of considering metabolic factors to combat drug resistance. It highlights the different challenges and metabolic alterations in non-small-cell lung cancer and small-cell lung cancer, underlining the need for subtype-specific treatments. Key signaling pathways, including PI3K/AKT/mTOR, MAPK, and AMPK, have been discussed for their roles in promoting drug resistance and metabolic changes, alongside the complex regulatory networks involved. This review article evaluates emerging treatments targeting metabolism, such as metabolic inhibitors, dietary management, and combination therapies, assessing their potential and challenges. It concludes with insights into the role of precision medicine and metabolic biomarkers in crafting personalized lung cancer treatments, advocating for metabolic targeting as a promising approach to enhance treatment efficacy and overcome drug resistance. This review underscores ongoing advancements and hurdles in integrating metabolic considerations into lung cancer therapy strategies.

## 1. Introduction

Cancer refers to a group of diseases characterized by the uncontrolled growth and malignancy of cells in the body. Lung cancer is among the top five most highly diagnosed cancers in the world and the second most diagnosed cancer in both men and women, as well as being the most common cause of death due to cancer. The very high mortality rate also surpasses the death occurring due to breast, colorectal, and prostate cancers combined. The primary cause of this illness is cigarette smoking, which accounts for about 80% of instances of lung cancer [1]. Lung cancer has a very high incidence rate, with an estimated 238,340 new cases (117,550 men and 120,790 women) expected be diagnosed and 127,070 deaths projected to occur due to the disease. In India, lung cancer accounts for 5.9% of all cancers and 8.1% of all cancer-related deaths [2]. As per a report, approximately 6.2% of men and 5.8% of women in the general population have the propensity to develop lung cancer in their lifetime. This means that 1 in 16 men and 1 in 17 women are at risk of developing lung cancer during their lifetime, with smokers and ex-smokers being at higher risk [3]. Most of the cases of lung cancer are diagnosed at later stages, rendering the treatment ineffective, and thus the disease becomes typically lethal. However, in the past twenty years, exciting gains in survival rate have resulted from the advancement in the screening techniques and their implementation, development of tailored treatments, and growing understanding of cancer biology. Primarily two types of lung cancer are known, non-small-cell lung cancer (NSCLC), which accounts for 80% of cases, and small-cell lung cancer (SCLC), which accounts for 15% of cases. NSCLC is further categorized as adenocarcinoma, squamous cell carcinoma, and large-cell carcinoma [4]. Adenocarcinoma mainly originates in the glands that secret mucus, whereas squamous cell carcinoma is mainly seen in the epithelium lining of lungs. In contrast to adenocarcinoma, squamous cell carcinoma is more aggressive and develops from the cell lining of the airway in the lungs. In contrast to the other two NSCLC subtypes, large-cell carcinoma is more aggressive and can arise from any part of the lung [5]. Small-cell carcinoma is the most prevalent subtype of lung cancer, followed by mixed small-cell carcinoma. SCLC is typically more aggressive than NSCLC and is slightly more prevalent in women (14%) than in males (13%). Metastasis is observed more frequently in SCLC patients than NSCLC patients, with the disease migrating outside the lungs in 94% of SCLC cases versus 70% of NSCLC cases [6].

Figure 1 depicts the significant advancements in lung cancer treatment. Despite these advances in therapeutics, resistance to drugs remains a major obstacle to long-term treatment effectiveness, eventually leading to therapeutic insensitivity, poor progression-free survival, and disease relapse [7]. Drug resistance refers to the reduction in the effectiveness of medication that occurs when cancers evolve and acquire resistance to the drugs used for treatment. Major treatment options for different types of lung cancer include surgery, chemotherapy, radiotherapy, immunotherapy, targeted therapy, etc. Major issues related to drug resistance are therapeutic insensitivity, poor progression-free survival, and disease relapse. Resistance mechanisms in lung cancer can arise from various factors, such as genetic mutations, epigenetic modifications, uncontrolled drug efflux, tumor hypoxia, changes in the tumor microenvironment, and several other cellular and molecular alterations [8].

Reprogramming of metabolic pathways is an essential characteristic of tumor cells. To meet the material, energy, and redox force requirements for fast multiplication, tumor cells remodel their metabolic pathways. Metabolic reprogramming, achieved through alterations in gene expression, cell state, and the tumor itself, causes certain metabolites to change in quantity or type both inside and outside of cells, generating an environment that is conducive for tumor growth and progression [9]. The various metabolic changes that take place in tumor cells include alteration in glucose metabolism, amino acid metabolism, and lipid metabolism. This reprogramming of tumor cells is beneficial for conferring drug resistance to many cancer cells [10,11,12].

Accordingly, this review aims to provide an overview of the currently available therapeutic approaches to treat lung cancer, as well as issues pertaining to drug resistance that persist despite prolonged treatment. This review also explores drug resistance signaling pathways and the role of tumor cell metabolic reprogramming in developing drug resistance in the tumor microenvironment. In addition, this work explores the junction of medication resistance and metabolic reprogramming to identify and create novel and effective therapeutic approaches for this persistent problem.

**Figure 1 diseases-12-00093-f001:**
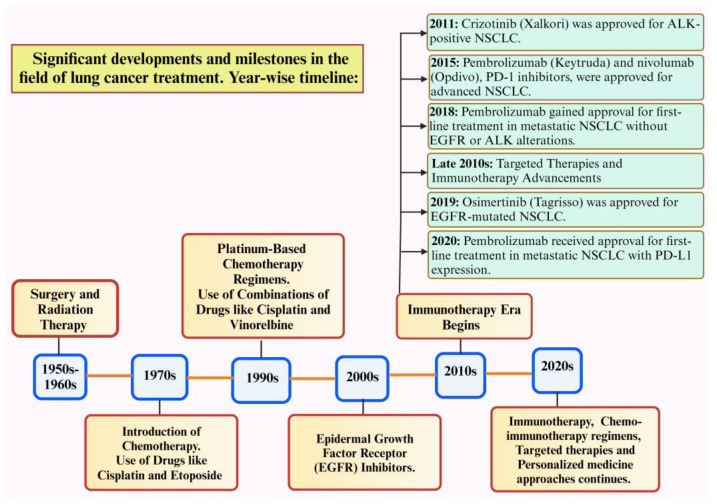
Timeline for the discovery of significant advancements in lung cancer treatment [13]. Created with Bio Render.

## 2. Signaling Pathways Involved in Lung Cancer

Figure 2 illustrates the various signaling pathways in the lung, where aberrations at different levels are implicated in lung cancer development and progression. The phosphatidylinositol 3-kinase (PI3K)/Akt pathway is pivotal in regulating diverse cellular processes and is frequently dysregulated in cancers, playing a significant role in the initiation and advancement of tumors. The mitogen-activated protein kinases/extracellular signal-regulated kinase or the RAS/RAF/MEK/ERK (MAPK) predominantly engages in apoptosis, pathogenesis, progression survival, and invasion. Further, there are other intracellular signaling pathways in the lung that, when downregulated, lead to cancer and serve as important targets for therapeutic treatments. These include the DNA repair pathways, inflammatory signaling pathway, anaplastic lymphoma kinase (ALK) pathway, and tumor suppressor genes. Additionally, these signaling pathways interact with one another; thereby, enhancing one pathway may either strengthen or inhibit another [14].

The PI3K/AKT/mTOR signaling pathway has cross-talk with the Ras/Raf/MAPK (MEK)/ERK pathway and tumor suppressor gene pathway: PI3K, a family of lipid kinases and class IA PI3Ks, which comprises a catalytic p110 subunit and a regulatory p85 subunit. Upon ligand binding to receptor tyrosine kinases (RTK) on the cell membrane, such as epidermal growth factor receptors (EGFRs), vascular endothelial growth factor receptors (VEGFRs), and others, the PI3Ks are activated, leading to the phosphorylation of PIP2 to PIP3 (phosphatidylinositol (4,5)–bisphosphate to phosphatidylinositol (3,4,5)–trisphosphate) catalyzed by the p110 subunit. The phosphatase and tensin homolog (PTEN) convert PIP3 to PIP2, suppressing the PIP3-dependent processes and thus restricting cell survival, growth, and proliferation. Next, phosphoinositide-dependent protein kinase (PDK1) and mTORC1 (a serine/threonine kinase) mediate the phosphorylation and activation of Akt together. It is known that the kinase mTOR is negatively regulated by the TSC1/TSC2 complex (tuberous sclerosis protein 1/2), which inhibits Ras homolog enriched in brain (Rheb) activation, a GTPase that activates mTOR. Accordingly, activated AKT kinases can phosphorylate both TSC1 and TSC2. Akt also inhibits the pro-apoptotic Bcl-2 protein family members and indirectly upregulates p53 by activating mouse double minute 2 homolog (MDM2) and NF-κB transcription factors, leading to increased expression of cell survival and anti-apoptotic signals. Dysregulation of the PI3K/Akt/mTOR pathway has been found to be involved in tumorigenesis and disease progression in NSCLC, including angiogenesis, migration, metastasis, proliferation, and apoptosis [15,16]. The VEGF/VEGFR-2 dimers mediate tumor metastasis, angiogenesis, and tumor survival through two major signaling pathways branches: Ras/MAPK and PI3K/Akt. Upon VEGF binding to VEGFR, dimerization and phosphorylation of the intracellular tyrosine kinase domain occur, followed by the activation of downstream proteins such as proto-oncogene tyrosine-protein kinase Src (c-Src), Shc-like protein (Sck), and VEGF receptor-associated protein (VRAP). Activation of c-Src involves the adaptor molecule growth factor receptor-bound protein 2 (GRB2)-associated binder 1. The adaptor proteins like GRB2 and SOS (Son of Sevenless) then recruit Ras (which is a type of small GTP-binding protein) and phosphatidylinositol 3-kinase (PI3K), leading to the formation of the signaling pathway branches Ras/MAPK and PI3K/Akt [17].

The mitogen-activated protein kinase/extracellular signal-regulated kinase (MAPK (MEK)/ERK) pathway is also known as the Ras/Raf/MAPK (MEK)/ERK pathway. After membrane receptor activation, adaptor proteins such as SOS recruit Ras proteins to activate Raf, a serine/threonine protein kinase, which becomes phosphorylated and activates MAPK kinase (MEK). This activation is followed by MEK phosphorylation and subsequent activation of ERK (extracellular signal-regulated kinase). ERK then activates its cytoplasmic and/or nuclear targets. Under normal conditions and in the cancer cell, the PI3K/AKT pathway also interacts with the MAPK/ERK node [17,18].

***Inflammatory signaling pathway:*** Tumor necrosis factor (TNF) binds and activates TNF receptors TNFR1 or TNFR2 that, in turn, activate the inflammatory signaling pathway. Both the cytokine and its receptors TNFR1 and TNFR2 are expressed in lung cancer. TNF stimulates the activation of nuclear factor-κB (NF-κB), which is triggered by activation of inhibitor of NF-κB (IκB) kinase (IKK). NF-κB translocates into the nucleus and promotes transcription of many NF-κB target genes, many of which (such as FLIP: cellular FLICE-inhibitory protein, BCL-XL: B-cell lymphoma-extra-large, and others) have anti-apoptosis and pro-survival function [19].

***DNA repair pathway:*** The tumor suppressor proteins, namely breast cancer-associated proteins BRCA1 and BRCA2, play an important role in DNA repair through a specific type of DNA repair pathway called double-strand break repair by homologous recombination (HR) repair. Many other genes, known as BRCA-related genes, also participate directly or indirectly. Mutations in the BRCA1/2 genes have been observed in 5–10% of NSCLC cases [20]. BRCA1 and BRCA2 regulate HR repair by facilitating the assembly of the DNA recombinase-RAD51 onto broken DNA ends at the site of double strand breaks (DSBs) and holding up replication forks [21].

The anaplastic lymphoma kinase (ALK) pathway: Fusion of ALK with other genes leads to constitutive ALK activation, promoting cell proliferation and survival through downstream signaling pathways, including PI3K/AKT and MAPK/ERK [22].

## 3. Mechanisms of Drug Resistance and Metabolic Reprogramming

### 3.1. Molecular Basis of Drug Resistance

One of the significant issues that reduces the efficacy of chemotherapies employed to treat cancer is drug resistance. Prior to treatment, tumors may exhibit innate resistance to chemotherapy. However, tumors that are initially responsive to chemotherapy may potentially develop drug resistance during treatment. It is estimated that more than 90% of patients with metastatic cancer experience treatment failure due to resistance to chemotherapy. Additionally, resistant micro metastatic tumor cells may also diminish the efficacy of chemotherapy in the adjuvant setting [23]. The molecular analysis of several of the most frequently mutated molecules in lung cancer includes V-Ki-ras2 Kirsten rat sarcoma viral oncogene homolog (KRAS; 25%), anaplastic lymphoma kinase (ALK; 6%), epidermal growth factor receptor (EGFR), phosphoinositide-3-kinase, catalytic, alpha polypeptide (PIK3CA; 3%), v-Raf murine sarcoma viral oncogene homolog B1 (BRAF; 3%), and human epidermal growth factor receptor 2 (HER2; 1%) [24].

### 3.2. Metabolic Pathways and Their Alterations in Lung Cancer

Alterations in major metabolic pathways in signaling pathways involved in lung cancer progression include glycolysis, amino acid metabolic pathways (such as glutamine, serine, glycine, tryptophan), and fatty acid synthesis pathways. Metabolic reprogramming or alteration is due to downregulation or upregulation of different metabolite levels in tumor cells compared to normal cells [25,26]. Understanding these metabolic abnormalities and identifying relevant metabolites that differ from normal metabolites could facilitate the implementation and evaluation of novel treatment strategies and targeted medicines for patients with lung cancer (LC). Changes in metabolomic pathways were verified by quantitative proteomics analysis of the main enzymes found in the disrupted pathways. This analysis identified 13 distinct biomarkers linked to metabolic disruption of NSCLC morbidity, implicated in 4 major pathways: aminoacyl-tRNA biosynthesis, glycine serine and threonine metabolism, tyrosine metabolism, and sphingolipid metabolism. Important enzymes involved in these pathways, including phosphoserine phosphatase, argininosuccinic acid catenase, tyrosinase, and 3-phosphoglycerate dehydrogenase, showed distinct variations in expression, according to the proteomics study [27].

An increase in the levels of glucose transporters (GLUT) in tumor cells is responsible for altered metabolism. Both GLUT3 and GLUT5 are found to be higher in tumor cells. The elevated glucose levels inside cells lead to metabolic changes [28]. The Warberg effect is evident in lung cancer cells, where glucose is metabolized via lactic acid fermentation rather than being broken down to pyruvate as in normal cells. Tumor M2-Pyruvate Kinase (PKM2), a glycolytic enzyme, is detected to be increased in LC and several other malignancies, and it is necessary for the malignant transformation [29]. Lactate formed inside the cell is transported with the help of transporters called monocarboxylate transporters (MCTs). This lactate shuttle, mainly via MCT1 and MCT4, is an important mechanism through which cancer tissue maintains the balance of the connections between glycolytic and oxidative cells [30]. 

Lung cancer cells show an increased uptake of glutamine. After entering the cell, glutamine is converted into glutamate, which is further converted to α-ketoglutarate with the help of a mitochondrial enzyme glutaminase (GLS). This pathway is called glutaminolysis. The alanine-serine-cysteine-transporter-2 (ASCT2 or SLC1A5) mediates the uptake of these glutamines into the cancer cells, and it has been noted that LC patients express these transporters with greater frequency [31]. L-type amino acid transporter 1 (LAT1), also known as SLC7A5/SLC3A2, is one of the various types of transporters found in LC patients, and it plays a role of exchanging glutamine for vital amino acids like phenylalanine, valine, and methionine [32]. Another transporter, SLC38A3, controls the transfer of glutamine and histidine, which, in turn, promotes the metastasis of NSCLC by modulating the PDK1/AKT signaling cascade. HIF-1α activates SLC38A1 and provides a variety of mechanisms to alter the glutamine level in the cell, whereas HIF-2α controls and upregulates the expression of SLC1A5 [33,34]. 

The serine synthesis pathway is a crucial pathway for energy production in cancer cells, along with glycolysis. The precursors, namely 3-phosphoglycerate and glutamate, which are generated by the glycolysis and glutaminolysis pathways, respectively, drive the manufacturing of serine. Glycine produced during the glycolysis process can be used as a source of carbon for one-carbon metabolism [35]. An enzyme known as phosphoserine aminotransferase (PSAT1) helps oxidize and catalyze the precursor 3-phosphoglycerate into 3-phosphoserine and α-KG (alpha- ketoglutarate), which is subsequently transformed into serine with the aid of another enzyme known as 1-3-phosphosrine phosphatase [36]. It was found that nuclear factor erythroid 2–related factor 2 (NRF2) controls the expression of the critical serine synthesis enzymes phosphoglycerate dehydrogenase (PHGDH), PSAT1, and serine hydroxymethyltransferase-2 (SHMT2) by activating activating transcription factor 4 (ATF4) to promote glutathione and nucleotide formation. PSAT1 overexpression worsens the prognosis of cancer by encouraging cancer cell proliferation, metastasis, and chemoresistance [37].

Glycine decarboxylase is necessary for tumor-initiating cells. When glycine decarboxylase is inhibited in cells with elevated levels of serine hydroxy-methyltransferase 2 (SHMT2), the accumulation of glycine is converted into toxic compounds such as methylglyoxal and aminoacetone, resulting in cell growth arrest [38]. 

The fatty acid synthesis pathways are also affected in lung cancer cells. The de novo production of fatty acids requires the enzyme fatty acid synthase (FASN), which has been shown to be highly expressed in NSCLC patients. Additionally, FASN was discovered to play a role in the metabolism of glucose by inhibiting the AKT/ERK pathway, which modifies LC cells [39]. There are other enzymes upregulated in different types of lung cancer cells, such as ATP citrate lyase enzyme (ACLY) and AcCoA carboxylase (ACC1/2) [40]. Cholesterol synthesis is also affected, as reported in an in vitro study showing that the presence of 25-Hydroxycholesterol increased LC cell growth and invasion. A vital regulator of lung development and morphogenesis, thyroid transcription factor 1 (TTF-1), specifically targets ATP-binding cassette transporter A1 (ABCA1) in LC cells. TTF-1 may be utilized as a diagnostic marker for LC, as it was found to be overexpressed in LC samples [41].

## 4. Current Therapeutic Approaches and Metabolic Considerations

### 4.1. Overview of Current Lung Cancer Therapies

There are some differences in the treatment approach of both types of lung cancer, namely NSCLC and SCLC. SCLC is the more aggressive form of lung cancer, often diagnosed at a later stage with frequent metastases. Therefore, the primary treatment modalities used for SCLC typically include chemotherapy, radiotherapy, targeted therapy, and immunotherapy. Surgery is very rarely used in SCLC cases due to its aggressive nature. In contrast, for NSCLC, treatment options may include chemotherapy, chemoradiotherapy, radiotherapy, targeted therapy, immunotherapy, as well as surgery, depending on factors such as the stage and characteristics of the tumor.

#### 4.1.1. Chemotherapy

Chemotherapy consistently serves as the first line of treatment for cancer, representing a standard therapeutic approach for lung cancer. Chemotherapeutic drugs are broadly classified into four categories: (1) alkylating agents, such as platinum compounds like cisplatin and carboplatin, (2) microtubule-targeting drugs, including vinorelbine, paclitaxel, and docetaxel, (3) antimetabolites, exemplified by pemetrexed and gemcitabine, and (4) topoisomerase inhibitors like etoposide [42]. Notably, in NSCLC, the principal chemotherapy components comprise gemcitabine, taxanes, pemetrexed, cisplatin, and carboplatin. Additionally, targeted therapy medications like EGFR inhibitors (e.g., erlotinib) and VEGFR inhibitors (e.g., bevacizumab) are utilized. The mechanisms of action for these drugs vary: cisplatin, carboplatin, and gemcitabine disrupt the DNA repair system, induce DNA damage, and prompt programmed cell death (apoptosis) in the cancer cell [43], while taxane-based drugs disrupt microtubule dynamics, induce cell cycle arrest, and trigger apoptosis [44]. Conversely, pemetrexed and methotrexate, classified as antifolate drugs, can lead to cell cycle arrest in the S phase [45]. In the management of metastatic SLCL, the first line of treatment includes a combination of platinum and etoposide; nevertheless, it shows a very poor survival rate, as it relapses within one year [46]. Currently, topotecan stands as the sole medication conforming to the accepted standard of care for second-line treatment of SCLC [47].

#### 4.1.2. Radiotherapy

Radiotherapy is commonly utilized in combination with chemotherapy, especially in stage III unresectable NSCLC. The adverse effects of radiation therapy have decreased due to the development of intensity-modulated radiotherapy (IMRT), stereotactic body radiation (SBRT), and four-dimensional computed tomography (4DCT) [48]. Primarily, the mode of action of radiotherapy stems from DNA damage, which can elicit immune responses in lung cancer. As a result, the combination of radiotherapy and immunotherapy could improve the recovery in lung cancer patients [49]. Comparatively to conventional chemoradiotherapy, a phase 3 trial suggests that platinum-based chemotherapy, radiotherapy, and durvalumab (an immune checkpoint inhibitor of PD-L1) as combinational therapy for lung cancer offer greater benefits, potentially extending overall survival (up to 4 years) in patients with stage III NSCLC [50]. In the case of SCLC patients, radiotherapy in combination with chemotherapy is applicable in the initial stages of treatment but not suitable for later metastatic stages. Around 30% of newly diagnosed SCLC cases receive standard treatment with curative intent, involving four cycles of platinum-doublet chemotherapy combined with radiotherapy [51].

#### 4.1.3. Targeted Therapy

In lung cancer, targeted therapy refers to the application of medications or other compounds that specifically target and obstruct the growth and metastasis of cancer cells. These treatments function by either focusing on certain chemicals implicated in the initiation and spread of cancer or by obstructing the signals that propel the growth of cancer cells. Anaplastic lymphoma kinase (ALK), epidermal growth factor receptor (EGFR), ROS proto-oncogene 1 (ROS1), and serine/threonine-protein kinase B-Raf (BRAF) inhibitors have become key components of lung cancer therapy. Recently, a new class of medications has emerged that can target the formerly “un-druggable” KRAS mutation [52,53]. FDA-approved drugs for targeted therapy include gefitinib, erlotinib, afatinib, dacomitinib, and osimertinib, which are used as EGFR inhibitors. ALK inhibitors include crizotinib, alectinib, brigatinib, ceritinib, and lorlatinib. ROS1 inhibitors comprise crizotinib, lorlatinib, entrectinib, and brigatinib. RET (rearranged during transfection) inhibitors such as pralsetinib and selpercatinib are available [54,55]. The FDA-approved medications have significantly improved patients’ overall survival (OS). For instance, gefitinib has improved median progression-free survival (mPFS) by approximately 10.8 months, erlotinib by approximately 14 months, afatinib by approximately 48 months, and dacomitinib by up to 14.7 months [56,57]. Some medications are currently undergoing clinical trials. For example, pyrotinib has demonstrated an extension of patients’ PFS by 6.9 months and median OS by 14.4 months in patients with advanced NSCLC with HER2 mutation [58]. Combination treatments for lung cancer, such as bevacizumab, a monoclonal antibody that targets VEGF, and erlotinib, an EGFR inhibitor, can extend the PFS in patients with NSCLC. In advanced NSCLC with an EGFR mutation, the combination of apatinib, a VEGFR inhibitor, with gefitinib, a first-generation EGFR tyrosine kinase inhibitor (TKI), may extend the mPFS for 19.2 months. However, this combination therapy has some adverse effects and lowers quality of life [59].

#### 4.1.4. Immunotherapy

Immunotherapy represents a novel therapeutic approach for cancer treatment and holds significant appeal to researchers due to its promising result in dealing with the disease. It entails boosting the immune system’s capacity to identify and combat cancerous cells. Even while cancer cells frequently find ways to avoid detection, the immune system can identify and eliminate aberrant cells, including cancer cells. The goal of immunotherapy is to get past these defense mechanisms and improve the immune system’s capacity to recognize and destroy cancer cells [60]. This therapy mainly focuses on inhibition of immune checkpoints cytotoxic T-lymphocyte–associated antigen 4 (CTLA-4), programmed death 1 (PD-1), and programmed death-ligand 1(PD-L1). These antibodies demonstrate potential in restoring antitumor immunity by targeting the host immune system to compel it to combat cancer cells [61]. Currently, FDA-approved immunotherapies in use include adoptive T-cell therapy, cancer vaccine, immune checkpoint inhibitors (ICIs), and cytokine modulators. Among the monoclonal antibodies utilized for treating lung cancer are anti-CTLA-4 antibody ipilimumab, anti-PD-1 antibodies pembrolizumab and nivolumab, and anti-PD-L1 antibodies atezolizumab, durvalumab, and avelumab [62]. The efficacy and overall survival outcomes of these drugs vary among patients based on their mutation. Studies have demonstrated that in metastatic NSCLC patients with high PD-L1 expression (≥50%) that lack EGFR or ALK mutation, drugs such as pembrolizumab and atezolizumab could improve PFS for approximately 10.3 months and median OS for 15.5 months [63]. In a phase III trial, durvalumab was shown to improve PFS up to 907 days and OS up to 1,420 days in patients with unresectable stage III NSCLC following failed chemotherapy and radiotherapy [64]. Immunotherapy is also used in combinational therapy along with chemotherapy and radiotherapy. Ongoing trials suggest an improvement in survival rates when immunotherapy is combined with other monoclonal antibodies, chemotherapy, or radiotherapy. For example, durvalumab, an anti-PD-L1 antibody, demonstrates better outcomes when used in combination rather than alone. One of the most prominent combinations involves radiation or chemotherapy along with an anti-CTLA-4 antibody called tremelimumab and durvalumab [64]. A phase 3 trial shows that when pembrolizumab, an anti-PD-1 antibody, is used with radiotherapy, it can enhance the overall survival of metastatic NSCLC patients [65]. Further, immunotherapy is used as a first-line treatment for SCLC in combination with chemotherapy, which also yields good outcomes in SCLC patients.

#### 4.1.5. Antiangiogenic Therapy

Angiogenesis is a physiological process by which new blood vessels are formed. In cancer cells, there is a significant increase in angiogenesis. Within the tumor microenvironment, as cells proliferate, they require more blood vessels to supply blood, leading to an increase in this process. The major angiogenic factors are vascular endothelial growth factor (VEGF), fibroblast growth factor (FGF), tumor necrosis factor-alpha (TNF-α), transforming growth factor-beta (TGF-β), and angiopoietins (Ang) [66]. In the tumor microenvironment, VEGF may suppress the immune cells’ response, and, consequently, VEGF inhibitors may also boost immune cell capability. Therefore, antiangiogenic therapy could be combined with immunotherapy to benefit cancer patients [67]. The FDA has approved bevacizumab, a monoclonal antibody that targets VEGF, for use in the treatment of NSCLC. Bevacizumab, when combined with monoclonal antibodies such as atezolizumab, has been confirmed as a potential therapy for non-squamous NSCLC with higher PD-L1 expression (≥50%) but without EGFR/ALK/ROS1 mutations. In patients of NSCLC carrying KRAS and STK11 mutations, as well as Serine/Threonine Kinase 11 (STK11), Kelch-like ECH-associated protein 1 (KEAP1), tumor protein p53 (TP53), and/or strong PD-L1 expression (≥50%), bevacizumab in combination with atezolizumab and chemotherapy (carboplatin and paclitaxel) could serve as the first-line treatment [68].

### 4.2. Impact on Metabolic Pathways

Lung cancer therapeutic approaches have a significant effect on metabolic pathways of cells. Metabolism refers to the complex set of chemical processes that occurs within living organisms to maintain life. In recent time, various metabolomics studies show the difference in metabolic pathways in patients undergoing treatment. Studies have been conducted to investigate the proteomic and genomic effects of the chemotherapeutic drug anlotinib. Anlotinib is a novel oral tyrosine kinase inhibitor that targets platelet-derived growth factor receptor-α, c-Kit, fibroblast growth factor receptors 1, 2, and 3 (FGFR), and vascular endothelial growth factor receptors 1, 2, and 3 (VEGF) and suppresses the proliferation of cancer cells [69]. Anlotinib regulates some essential cancer cell metabolism like amino acids metabolism of tryptophan, threonine, glycine, serine, and phenylalanine and amino acid biosynthesis of valine, leucine, isoleucine pathways, and glyoxylate and decarboxylate pathways [70]. Additionally, certain metabolites serve as potential biomarkers to assess the efficacy of anlotinib, such as glycine-associated glycocholic and glycodeoxycholic acid [71]. Some chemotherapeutic drugs show alteration in metabolism, as it leads to a reduction in valine and lactate, ultimately showing relieved glycolysis. Drugs show their effect by reversing the Warburg effect and also affecting the high serum level of lipids, choline, and isobutyrate, etc. A549 cells (adenocarcinomic) treated with cisplatin have increased lipid levels but decreased niacinamide levels [72]. Chemotherapy causes the integrity of the cell membrane to be disrupted, which leads to cell membrane rupture and degradation. Lipids and glycoproteins are consequently more concentrated in the blood. Additionally, it has been proposed that chemotherapy raises the levels of the 3-hydroxybutyrate metabolite in sera, which, in turn, enhances lipolysis [73]. Chemotherapy is necessary for many biological functions. It alters energy metabolism, disrupts the production of phosphatidylcholine, disrupts the metabolites in lung cancer serum, and has an impact on DNA replication and microtubule function. Therefore, any chemotherapy treatment may disrupt metabolic pathways and result in the production of metabolites [74]. Patients with lung cancer are expected to have altered metabolite profiles following surgery. Metabolomic studies could offer a deeper understanding of these operation-related modifications, as a study showed higher levels of sphingolipids such as ceramide and sphingomyelin in lung cancer patients compared to controls in both pre- and post-surgery patients [75].

### 4.3. Considerations for Overcoming Drug Resistance

Drug resistance in lung cancer is a significant challenge in the management of the disease. It refers to the ability of cancer cells to survive and continue growing despite the presence of drugs designed to eliminate or control them. There are various mechanisms through which lung cancer cells can develop resistance to different types of treatment, including chemotherapy, targeted therapy, and immunotherapy. Lung cancer exhibits drug resistance through various key mechanisms, including multidrug resistance (MDR), DNA repair mechanisms, activation of alternative pathways, changes in tumor microenvironment, intertumoral heterogeneity, etc. [76]. To address these challenges, targeted and multifunctional nanoscale drug constructions have been developed as a result of nanotechnology advancements. These constructs offer promising methods to overcome important biological barriers, such as encapsulating different active treatments, surface functionalization of nanomedicine components, and potential regulation of these components. These characteristics improve the transport of various therapeutic drugs directly to the tumor microenvironment (TME), which reverses the anticancer treatment-resistant state of LC [77]. Some of the newer nanoparticle-based approaches include applying magnetic nanoparticles (NPs), namely the commercially available ferrofluids, as intravenous medication delivery systems. After injection, the tumor site is exposed to an external magnetic field, which causes the NPs to accumulate there and decreases the drug’s systemic toxicity. Moreover, NPs have the potential to serve as carriers of several anticancer agents, such as radionuclides, cancer-specific antibodies, and genes. On the other hand, within the past 10 years, photodynamic therapy (PDT) has become a viable therapeutic strategy for the treatment of cancer. As most photosensitizers exhibit hydrophobic properties, suitable delivery mechanisms are necessary [78]. Delivering siRNAs via NPs is a further novel approach to stop DR. Different siRNAs could be delivered in tandem to silence many genes, including the DR-related genes. The physicochemical characteristics of siRNAs hinder their cellular uptake due to their difficulty in navigating phospholipid membranes. Thus, the creation of novel RNAi technologies and suitable carriers is necessary. The technology of genome editing could offer a foundation for the creation of fresher, more effective strategies; another recent approach is the use of biocompatible compounds, and the use of biopolymers, such as tamarind seed polysaccharide PST, to prepare Paclitaxel (PTX)-loaded NPs through epichlorohydrin crosslinking [79]. Using nanoparticles to deliver therapy to cure drug-resistant lung cancer is a growing area of research to deal with the drug resistance issue.

## 5. Emerging Therapeutic Strategies Targeting Metabolism

In addition to the present therapeutic approaches, there are many new emerging treatment options under development, which mainly target the metabolic changes that occur due to cancer progression. Targeting metabolism has been an area of interest in the development of therapeutic strategies for lung cancer. Some major metabolic processes that can be targeted are glucose metabolism inhibition, mitochondria, fatty acid, amino acid metabolism, targeting oncogenic pathways, and immunometabolism [80]. Exciting developments in molecular targeted therapy and immunotherapy in combination with new surgical, pathological, radiographical, and radiation techniques have resulted in spectacular improvement in patients’ survival. KRAS is nearly exclusively seen in adenocarcinomas, while TP53 is the most frequently mutated gene in non-small-cell lung cancers (NSCLCs). RAF-MEK-MAPK and PI3K-Akt-mTOR are the two signaling pathways that are most frequently altered in lung cancer [81]. In NSCLC tumors, TP53 is mutated in approximately 46% of cases. This mutation leads to reduced apoptosis, primarily due to the suppression of TP53-induced glycolysis and apoptosis regulator (TIGAR). Further, there is enhanced glycolysis and glucose transport attributed to the upregulation of phosphoglucomutase (PGM). Additionally, TP53 is no longer capable of inhibiting the activity of glucose 6 phosphate dehydrogenase (G6PD) [82]. In addition to controlling glucose levels, the PI3K-Akt-mTOR signal transduction pathway triggers cancerous mutations that have a variety of negative side effects, including angiogenesis, proliferation, and differentiation. As a result, it establishes a clear connection between metabolic targeting, treatment response, and carcinogenesis [80]. Most of the cancer signaling pathways mutated in lung cancer, like RAS-RAF-MEK-MAPK pathway, are mutated in 58% of all NSCLC tumor’s and 76% of lung adenocarcinomas [83], and KRAS and EGFR pathways, KEAP1/NRF2 pathway, etc., have a great impact on major metabolic pathways like glucose metabolism, amino acid metabolism pathways, and reactive oxygen species (ROS) regulation [84]. Therefore, proper targeting of these metabolic pathways gives us a better approach to develop new therapeutic options. There are various drugs under development that target glucose metabolism. Glucose transporter (GLUT) inhibitors like fasentin, WZB117 (2-fluoro-6-(m-hydroxybenzoyloxy) phenyl m-hydroxybenzoate), DRB18 (a pan-GLUT inhibitor), STF31 (selective glucose transporter GLUT1 inhibitor), etc., are under pre-clinical trials. These drugs are mainly targeting the transport of glucose. WZB117 and DRB18 demonstrated inhibition under in vitro and in vivo conditions in NSCLC cancer models [85]. Shikonin, a drug used for late-stage lung cancer treatment, is a pyruvate kinase M2 (PKM2) inhibitor that is given to patients not undergoing surgery, radiotherapy, or chemotherapy. Activation of pyruvate kinase M2 (PKM2) appears to be inactivated in cancer and raises the ratio of glycolysis to glucose oxidation [86]. Some other drugs for lung cancer therapy under trial include dichloroacetate (DCA), which is a pyruvate dehydrogenase kinase 1 (PDK1) inhibitor. Also, PSTMB targets lactate dehydrogenase A (LDHA) [87]. 

Major targets for mutation in lung cancer include EGFR mutations, ALK rearrangements, ROS1 rearrangements, RET rearrangements, neurotrophic tyrosine receptor kinase (NTRK) fusions, mesenchymal–epithelial transition factor (MET) exon 14 skipping mutation, KRAS G12C mutation, BRAF V600E mutation, and ERBB2 (HER2) mutation. FDA-approved drugs are available for targeting these mutations; still, further translational research is exploring novel driver mutations, converting traditionally undruggable mutations into therapeutic targets [88]. Several drugs are currently under phase 1/2 clinical trials with different mechanisms of action compared to existing drugs. For example, Patritumabderuxtecan is a monotherapy drug under phase 1 clinical trial; it is an anti-HER3 antibody–drug conjugate targeting mutated EGFR and NSCLC that has progressed after tyrosine kinase inhibitors (TKI) showing an objective response rate (ORR) of 39%. Other monotherapy drugs under trial include Pemigatinib (aFGFR1-3 inhibitor), Telisotuzumabvedotin (anti-MET antibody–drug conjugate), Taletrectinib (ROS1 inhibitor), and Trastuzumabderuxtecan (anti-HER2 antibody–drug conjugate) [89,90]. 

Combinational therapy is a recent approach that targets multiple pathways associated with cancer development, such as PI3K/AKT/mTOR and RAF/MEK/ERK, by focusing on tumor cell receptors. There are certain drugs under different phases of clinical trials, like Anlotinib (multi targeting TKI), and used in combination with Osimertinibis as a first line of therapy for mutated EGFR and NSCLC [91]. Other drugs under clinical trials are Alisertib (Aurora kinase inhibitor), Sapanisertib (mTOR inhibitor), and Repotrectinib (ROS1/TRK/ALK inhibitor); all of these are used in combination with Osimertinib [92,93]. Some of the current therapeutic approaches under development or in clinical trials are listed in Table 1 below. 

## 6. Future Perspectives: Navigating the Integrated Landscape

Despite numerous advancements in biomedical research and the development of newer technologies for diagnosis and treatment, the high mortality rate of lung cancer persists. Consequently, it is imperative to adopt better analysis methods to yield improved outcomes. One such newer approach is systems biology; the goal of systems biology analysis is to comprehend the characteristics of a particular system, which, in the context of cancer, may include xenografts, tumor cell lines, genetically altered mice, or primary cells taken from patients either before or after treatment. Many components generated through interaction between the systems, and experiments in consecutive repeating cycles, are integrated with theory, simulation, and mathematical modeling [94]. The system biology approach in lung cancer is utilized for studying novel biomarkers, therapeutic targets, and more. Significant progress in understanding lung cancer biology has been made using a systems biology approach, which integrates data generated from siRNA library screens with data obtained from clinically significant LC samples or mouse models utilizing gene expression data, DNA sequencing, comparative genomic hybridization (CGH) analysis, miRNA profiling, and protein or protein phosphorylation signaling network assessments. This has led to better prognoses and therapy prediction techniques for NSCLC [95]. Key steps involved in the systems biology approach include comprehensive measurement and quantification of changes at the protein, DNA, RNA, or miRNA and signaling levels. Further, the data from measurements are combined to create a comprehensive picture of the system in issue, along with the evaluation of their dynamic changes, which, in the context of tumors, refers to the development of a more malignant phenotype, as well as their reactions to targeted agents, CT, and radiation therapies, and modeling the system with the help of the integrated data [96]. Recent studies show how we can narrow down our study related to genes involved in lung cancer using a systems biology approach. For instance, the Cluster ONE plugin, a cytoscape software, was used to analyze and cluster networks, revealing seven genes (BRCA1-TP53-CASP3-PLK1-VEGFA-MDM2-CCNB1 and PLK1) involved in diseases development. This comprehensive gene information approach provides valuable insights into the roles of these genes in various diseases [97]. In another study aimed at developing effective new markers for the early detection of NSCLC, a highly accurate model was generated using the least absolute shrinkage and selection operator (LASSO) for predicting non-small-cell lung cancer (NSCLC) using gene expression data and interaction networks. Using The Cancer Genome Atlas Program (TCGA) and (Genotype-Tissue Expression) GTEx data, the differentially expressed genes (DEGs) in NSCLC in comparison with normal tissues were discovered [98]. The above-mentioned studies show the wide application of systems biology and other integrated approaches to deal with lung cancer.

Along with the application of the systems biology approach, we need to use various cutting-edge technologies to better handle the diseases. For superior diagnosis and treatment options, artificial intelligence (AI) and machine learning (ML) approaches for precision medicine, nanotechnology-based drug delivery systema, immunotherapy, combinational therapy, and some other new approaches can be used [99]. Establishing an operational flow based on AI models and medical management platforms with high-performance computing is essential for precision cancer genomics in clinical practice to comprehend massive biodata in cancer genomics. The fundamental approach of using AI tools in cancer includes AI models for mutational analysis, single-cell genomics and computational biology, text mining for cancer gene target identification, computational prediction of pathogenic variants of cancer susceptibility genes, and NVIDIA graphics processing units [100]. Nanotechnology-based drug delivery systems can significantly increase the availability of therapeutic drug on tumor site while reducing toxicity. Treatment for lung cancer often involves the use of certain nanomedicine materials, such as lipid, polymer, magnetic, and hafnium oxide nanoparticles [101]. In addition to these advancements, newer radiotherapy approaches like boron neutron capture therapy (BNCT), proton therapy, brachytherapy, and stereotactic body radiation therapy (SBRT) have been developed [102,103]. There are some other emerging therapies for lung cancer like cryoablation, photodynamic therapy, hyperthermia, etc. Cryoablation is a therapeutic approach where tumors are destroyed by lowering the temperature. This procedure entails attaching cryoprobes to pressurized argon, causing the probe to rapidly cool, forming an ice ball at the tip of the cryoprobe. The freezing and thawing process damages the cell membrane and causes microvascular damage, leading to hypotonic stress and ultimately resulting in cell necrosis [104].

After cancer treatment, metabolic alterations in lung tumors include decreased glucose uptake, disturbed mitochondrial function affecting ATP synthesis, changes in amino acid and lipid metabolism, and the accumulation of hazardous metabolic byproducts. These alterations can potentially slow tumor growth and indicate the success of therapy, thereby identifying important targets for therapeutic approaches [105]. In photodynamic therapy, non-invasive photosensitive compounds and light activation that destroy the tumor region have been utilized. This technique has shown good efficacy in enhancing the rate of survival of lung cancer patients [106]. Hyperthermia therapy causes an elevation in the body tissue temperature. For this, heat is applied locally to raise the temperature to between 42 and 45 °C using external sources such microwaves, radio waves, lasers, ultrasound, etc. The goal of this procedure is to eradicate cancer cells or stop their proliferation without endangering healthy tissues [107,108]. 

Although there are many emerging therapeutic approaches, it is imperative to address all the above-mentioned challenges and limitations to develop a complete cure to minimize side effects and enhance efficacy of treatment in lung cancer. One of the major challenges is to make cost-effective treatment available to all patients, thereby reducing cancer-related mortality.

## 7. Conclusions

The fight against lung cancer, a leading global health challenge, necessitates a multifaceted approach due to the complex interplay of drug resistance, metabolic reprogramming, and the genetic landscape of the disease. This review has highlighted the critical role of targeting pathways, mutations, and genetic alterations, alongside the importance of understanding cancer metabolism for effective therapeutic strategies. The advancements in targeted therapies, immunotherapies, and combinational treatments, complemented by emerging options like metabolic inhibitors and nanocarriers, underscore the potential to significantly improve patient outcomes. Moreover, the integration of mathematical modeling and systems biology approaches offers promising avenues for enhanced detection and personalized treatment strategies. The adoption of AI-ML technologies and innovative techniques such as hyperthermia, cryoablation, and photodynamic therapy further expands our arsenal against lung cancer. Through a comprehensive understanding of the disease’s underlying mechanisms and leveraging interdisciplinary approaches and cutting-edge technologies, we can pave the way for more effective treatments and ultimately improve the survival rates and quality of life for lung cancer patients.

## Figures and Tables

**Figure 2 diseases-12-00093-f002:**
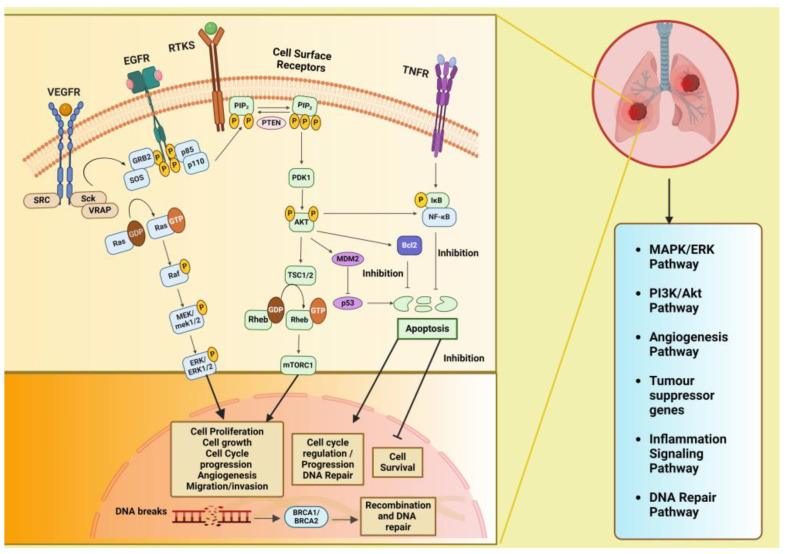
Signaling pathways involved in lung cancer that have a role in metabolic reprogramming viz. PI3K/Akt pathway, MAPK/ERK pathway, inflammation signaling pathways, tumor suppressor genes, and DNA repair pathways and their cross-talk. Created with BioRender.com. *Legends:* EGFR: epidermal growth factor receptor; VEGFR: vascular endothelial growth factor receptor, RTK: receptor tyrosine kinase; SRC: proto-oncogene tyrosine-protein kinase Src; Sck: Shc-like protein; VRAP: VEGF receptor-associated protein; A-Raf: serine/threonine-protein kinase A-Raf; Ras: Rat sarcoma gene; MAPK/ERK: mitogen-activated protein kinase/extracellular signal-regulated kinase; PI3K: phosphatidylinositol 3-kinase; PTEN: phosphatase and tensin homolog; PIP2: phosphatidylinositol bisphosphate; PIP3: phosphatidylinositol triphosphate; PDK1: phosphoinositide-dependent kinase 1: Akt: protein kinase-B; TSC1: tuberous sclerosis complex 1: TSC2: tuberous sclerosis complex 2; Rheb: Ras homolog enriched in brain; mTOR: mammalian target of rapamycin; MDM2: mouse double minute 2 homolog; Bcl-2: B-cell lymphoma-2; TNF: tumor necrosis factor; TNFR1/2: TNF receptor 1/2; NF-κB: nuclear factor-κB; BRCA1 and BRCA2: breast cancer-associated protein receptors. Arrows represent activation, bars represent inhibition.

**Table 1 diseases-12-00093-t001:** Current therapeutic approaches for lung cancer treatment.

Target	Signaling Pathway/ Metabolic Process	Drug Name/Type	Action/Effect	Development Stage	References
Glucose Transport	Glycolysis	Fasentin, WZB117, DRB18, STF31	Inhibits glucose transporter (GLUT), inhibits LDHA, disrupting lactate production and glycolysis	Pre-clinical trials	[85]
Pyruvate Kinase M2 (PKM2)	Glycolysis/Warburg effect	Shikonin	PKM2 inhibitor, encourages metabolic shift from glycolysis to glucose oxidation	Use in later-stage treatment	[86]
Pyruvate Dehydrogenase Kinase 1 (PDK1)	Glycolysis and mitochondrial metabolism	Dichloroacetate (DCA)	Inhibits PDK1, promoting oxidative phosphorylation over glycolysis	Under trial	[87]
Lactate Dehydrogenase A (LDHA)	Glycolysis	PSTMB	Inhibits LDHA, reducing lactate production and potentially disrupting cancer cell metabolism	Under trial	[87]
EGFR mutations, ALK rearrangements, and others	EGFR/ALK signaling pathway	Erlotinib, Afatinib, Osimertinib, Gefitinib, Crizotinib, Ceritinib, Alectinib, Brigatinib and others	Targets specific genetic mutations in lung cancer	FDA approved for specific mutations	[88]
Mutated EGFR, NSCLC	EGFR pathway	Patritumabderuxtecan	Anti-HER3 antibody–drug conjugate	Phase 1 clinical trial	[77]
FGFR1-3, MET, ROS1, HER2	Various pathways	Pemigatinib, Telisotuzumabvedotin, Taletrectinib, Trastuzumabderuxtecan	Various monotherapy drugs targeting different pathways	Human phase 1/2 clinical trials	[89,90]
EGFR mutated NSCLC	PI3K/AKT/mTOR, RAF/MEK/ERK	Anlotinib + Osimertinib	Multi-targeting TKI in combination with EGFR inhibitor	Clinical trials	[91]
Various targets	PI3K/AKT/mTOR, RAF/MEK/ERK	Alisertib, Sapanisertib, Repotrectinib + Osimertinib	Targeting tumor cell receptors with Osimertinib	Clinical trials	[92,93]

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
