# Peer review of "Unravelling the Triad of Lung Cancer, Drug Resistance, and Metabolic Pathways"

_diseases, 2024, doi:10.3390/diseases12050093_

Round 1
Reviewer 1 Report
Comments and Suggestions for Authors
The manuscript „Unravelling the Triad of Lung Cancer, Drug Resistance, and Metabolic Pathways“ by Mohanty and coauthors gives an overview of signaling pathways in lung cancer, discusses therapy approaches that differ depending on lung cancer type and diverse resistance mechanisms. In their review, authors are also focusing on changes in metabolic pathways in lung cancer and argue about possible therapeutic strategies to target cancer metabolism with the aim to overcome drug resistance and sensitize lung cancer to conventional therapy. The topic is very interesting and potentially of great scientific interest. The review is written in a understandable and clear way, is nicely organized so it is very easy to follow and mostly examines recently published papers. Although the review offers self-explaining illustrations, the text boxes in the Figure 2 are too small and thus the size of the letters needs to be adjusted to make a Figure 2 more clear to the reader. Nevertheless, I believe that the review will contribute greatly to the lung cancer field.
Comments on the Quality of English LanguageThere are few language misspellings/typos in the text, so I recommend the authors to carefully read the manuscript and revise it before publishing. Therefore, minor editing of English language is required in the manuscript.
Author Response
As per the suggetion, we have made changes in the figure and updated new in the manuscript.
Thank you very much
Reviewer 2 Report
Comments and Suggestions for Authors
1. The background yellow color of figure 1 and 2 is too bright to easily read the words. Meanwhile, the authors need to increase the front size of word in the figure legend.
2. In the section 2, only the signalling pathways related with metabolism should be discussed.
3. Key references should be added in the timeline of advancement in lung cancer.
4. The post-therapy change of cancer tissue is recently reported, such as chemotherapy ( Li et al. DOI: 10.1016/j.canlet.2023.216583) and radiotherapy (Wang et al. DOI: 10.1002/smtd.202200570). Therefore, in the future perspectives part, it will be very interesting to have discussion about the metabolic change of lung tumor after treatment.
Comments on the Quality of English LanguageMinor editing of English language required
Author Response
- The background yellow color of figure 1 and 2 is too bright to easily read the words. Meanwhile, the authors need to increase the front size of word in the figure legend.
Answer- As per suggestion we have modified the figure.
- In the section 2, only the signalling pathways related with metabolism should be discussed.
Answer- As per suggestion we have checked and maintained the structure as per the requirement.
- Key references should be added in the timeline of advancement in lung cancer.
Answer- As per suggestion we have added reference.
- The post-therapy change of cancer tissue is recently reported, such as chemotherapy ( Li et al. DOI: 10.1016/j.canlet.2023.216583) and radiotherapy (Wang et al. DOI: 10.1002/smtd.202200570). Therefore, in the future perspectives part, it will be very interesting to discuss the metabolic change of lung tumors after treatment
Answer- As per suggestion we have added a reference related to the metabolic change of lung tumor after treatment
Reviewer 3 Report
Comments and Suggestions for Authors
Dear Editor and Authors,
It was my pleasure as a oncological thoracic surgeon to read this very informative and well-constructed review titled “Unravelling the Triad of Lung Cancer, Drug Resistance, and Metabolic Pathways” in which the authors present a quite thorough overview of the relationship of drug resistance and metabolic pathways in lung cancer.
This is a good work but the manuscript needs extensive language editing by a native speaker or a professional service because although relatively easy to read there are expression mainly errors! For example therapeutics versus therapeutic!, advancements versus advancement! or Lines 69 – 71 in the introduction make no sense and need to be re-written.
Apart from those language corrections there is no need for much improvement.
Thank you kindly for asking me to review this work.
Kind regards
Comments on the Quality of English LanguageNeeds some moderate language editing and revising.
Author Response
- Lines 69–71 in the introduction make no sense and need to be re-written.
Answer- Thank you very much for the detailed review. As per suggestion we have rewritten the lines 69-71 in the introduction and highlighted the same in the manuscript. Beside this, we also improved the English in the entire manuscript.
Thank you very much
Reviewer 4 Report
Comments and Suggestions for Authors
The battle against lung cancer, which is one of the most significant global health challenges, requires a multifaceted approach due to the complex interplay of drug resistance, metabolic reprogramming, and the genetic landscape of the disease. This review emphasizes the critical role of targeting pathways, mutations, and genetic alterations and understanding cancer metabolism to develop effective therapeutic strategies. The progress in targeted therapies, immunotherapies, and combination treatments, along with emerging options like metabolic inhibitors and nanocarriers, highlights the potential to significantly improve patient outcomes.
In the final section of the manuscript, the authors outlined future perspectives. They emphasized the importance of applying a systems biology approach, which requires utilizing various advanced technologies to manage diseases better. Artificial intelligence (AI) and machine learning (ML) techniques can achieve superior diagnosis and treatment options for precision medicine, nanotechnology-based drug delivery systems, immunotherapy, combinational therapy, and other innovative approaches.
Minor suggestions:
- When the Authors discuss metabolic pathways (rows 229 – 280), they could schematically show them together with the corresponding transporters on the cell membrane (which deliver reactants, e.g., glucose, or remove metabolic products).
- The authors could emphasize correlations between different metabolic pathways and correlations between biomarkers arising from metabolic pathways.
- The authors divided chemotherapy drugs based on their mechanism of action. They could schematically present each mechanism of action (or the corresponding reaction, for example, alkylation) and give an example of the structure of some drugs from each group.
- They could give an example of a blood count with appropriate biomarkers before chemotherapy and radiotherapy and changes in these parameters during therapy.
Author Response
Thank you for your constructive feedback
We appreciate the suggestion to include schematic representations of metabolic pathways. We also highlight the correlations between different metabolic pathways and associated biomarkers is indeed valuable. Besides this we also highlighted the chemotherapy outline topic as a section in the manuscript which help for the reader to have more information about chemotherapy.
Best